# Tau as a Biomarker of Neurodegeneration

**DOI:** 10.3390/ijms23137307

**Published:** 2022-06-30

**Authors:** Sarah Holper, Rosie Watson, Nawaf Yassi

**Affiliations:** 1Department of Medicine—The Royal Melbourne Hospital, University of Melbourne, Parkville 3050, Australia; watson.r@wehi.edu.au (R.W.); yassi.n@wehi.edu.au (N.Y.); 2Population Health and Immunity Division, The Walter and Eliza Hall Institute of Medical Research, Parkville 3052, Australia; 3Department of Neurology, Melbourne Brain Centre at The Royal Melbourne Hospital, University of Melbourne, Parkville 3050, Australia

**Keywords:** tau, neurodegeneration, Alzheimer’s disease, biomarker, cerebrospinal fluid, tauopathy

## Abstract

Less than 50 years since tau was first isolated from a porcine brain, its detection in femtolitre concentrations in biological fluids is revolutionizing the diagnosis of neurodegenerative diseases. This review highlights the molecular and technological advances that have catapulted tau from obscurity to the forefront of biomarker diagnostics. Comprehensive updates are provided describing the burgeoning clinical applications of tau as a biomarker of neurodegeneration. For the clinician, tau not only enhances diagnostic accuracy, but holds promise as a predictor of clinical progression, phenotype, and response to drug therapy. For patients living with neurodegenerative disorders, characterization of tau dysregulation could provide much-needed clarity to a notoriously murky diagnostic landscape.

## 1. Introduction

Few proteins in the central nervous system are as abundant and peculiar as tau. Discovered as recently as 1975, tau’s original characterization as an axon-specific intracellular neuronal protein with purely microtubule-related functions has proven inaccurate on all fronts. Under physiological conditions, tau exists in multiple neuronal compartments [1], is present at basal levels in the brain’s interstitial fluid [2], is expressed by several glial cell types [3], and has various non-microtubule-associated functions [4]. Tau’s natively flexible, disordered structure permits interactions with multiple binding partners, which lends it to such ubiquity and multifunctionality. An impressive repertoire of post-translational modifications offers tau further opportunity for function-specific adaptation. Alas, under pathological conditions, these same characteristics see tau readily accumulate abnormal modifications and bind with other tau proteins to form pathological aggregations. These complexes are toxic to brain cells, resulting in a phenotypically diverse range of neurodegenerative disorders dubbed tauopathies, with Alzheimer’s disease (AD) being the most prevalent.

Over a century since Alois Alzheimer’s first description of AD pathology at autopsy, a ‘definitive’ AD diagnosis still requires the post-mortem demonstration of neurofibrillary tangles (NFTs) and amyloid beta (Aβ) plaques [5]. In fact, all tauopathies require autopsy for definitive diagnosis, owing to the challenges of in vivo clinical diagnosis among this clinically heterogeneous group of disorders. Such a retrospective diagnosis is of course of no utility to patients living with neurodegenerative disorders, for whom a clear diagnostic label could offer valuable clarity and prognostic information for them and their family. 

The past decades have seen a revolution in the development of biomarkers: fluid- or imaging-based indicators of underlying pathology. Today, elevations in the total and phosphorylated tau concentrations in the cerebrospinal fluid (CSF), along with low Aβ42 levels, are central to the National Institute on Ageing and Alzheimer’s Association [6] and International Working Group [7] research criteria for an AD diagnosis.

In this review, we first provide a historical context for the subsequent discussion with a recap of the key discoveries in our understanding of tau in physiology and disease, and its development as a biomarker. Next, we consider tau’s expression and structure, and how this relates to tau’s function in the healthy nervous system. The processes by which tau may become deranged are then described, including a review of certain tauopathies and their clinical and pathological features. Tau’s role as a biomarker in biological fluids, including the CSF and blood-based fluids, is considered, with a focus on clinical applications offering much-needed diagnostic and prognostic information for patients living with neurodegenerative disorders. Challenges with the interpretation of these results are discussed, as well as possible solutions to these challenges such as the utility of combined biomarker analyses. Overall, this review highlights the remarkable advances in our understanding of tau as a biomarker of neurodegeneration in the 50 years since the molecule’s first isolation.

## 2. From Tangles to Tubules to Tau: Pivotal Discoveries Pertaining to Tau

### 2.1. NFTs Comprise Pathological Filaments

NFTs were first identified by Alois Alzheimer in his seminal 1907 paper: ‘On an Unusual Illness of the Cerebral Cortex’, a clinicopathological case report of the rapid dementing process of a 51-year-old woman admitted to the insane asylum of Frankfurt am Main [8]. On post-mortem examination of the woman’s brain, Alzheimer remarked on the ‘very striking’ appearance of intracellular fibrils ‘combined in thick bundles’. In sections of the woman’s brain demonstrating more advanced neurodegeneration, Alzheimer also described what would now be called ‘ghost tangles’ where ‘only a tangle of fibrils indicates the place where a neuron was previously located’.

Half a century later, the advent of commercial scanning electron microscopes allowed the NFT’s structural components to be discovered. In 1963, Kidd and Terry independently discovered that NFTs comprised filaments twisted into double helices alternating between 15 nm and 30 nm in width, with a cross-over span of 80 nm [9,10]. Kidd termed these structures ‘paired helical filaments’ (PHFs); Terry opted for ‘twisted tubules’; the former name took hold. A second less prevalent NFT component, the 15 nm wide rod-like straight filament (SF), lacking alternating width modulation, was discovered in 1968 [11].

The distinction between these two filamentous species became somewhat moot in 1991 with the recognition that they represent different arrangements of the same subunit [12]. Using electron diffraction techniques, Crowther identified that PHFs’ distinctive scalloped edges are generated by stacks of C-shaped subunits arranged base-to-base (i.e., forming an ‘S’ shape in cross section). The same subunits stacked back-to-back (i.e., generating an ‘X’ shape in cross-section) resulted in SFs. The ‘shared subunit’ theory was further strengthened by identical epitopes on antibody labelling, and Crowther’s identification of hybrid filaments: species of undulating width then a sharp transition to a straight segment.

NFTs form in a sequence of histopathologically distinct stages first described in 1989 [13]. NFT formation begins with the accumulation of abnormally phosphorylated tau in the cytoplasm of affected neurons. At the ultrastructural level, these pre-fibrillary deposits comprise PHFs and SFs. Several filaments may condense into small bundles, giving the cytoplasm a granular appearance under light microscopy. With further aggregation, the bundles form the rod-shaped inclusions characteristic of early (stage 1) tangles, often condensed near the nucleus and dendritic branch points. Progression to mature (stage 2) tangles involves the filaments becoming more densely packed, aggregating into large flame-shaped inclusions whose presence may be associated with a dislodged or shrunken nucleus. Ultimately the neuron dies to leave an end-stage tangle in its place. These stage 3 or ‘ghost’ tangles adopt a looser arrangement, existing freely in the neuropil.

### 2.2. Tau Is Discovered and Identified to Be the Abnormal Protein in NFTs

Investigations followed to clarify the biochemical composition of these NFTs. In 1974, a New York-based team including Terry, led by Iqbal, performed electrophoresis of NFTs isolated from post-mortem AD patients’ brains and identified that the major PHF protein was a unique 50 kDa protein [14]. In the absence of further defining features, this protein was simply called ‘PHF protein’. A year later, an entirely separate New Jersey-based research group led by Weingerten reported the isolation of a unique ‘heat stable protein essential for microtubule assembly’ from a porcine brain [15]. Weingerten named the protein tau, the Greek letter for T, ‘for its ability to induce tubule formation’. It took over a decade for it to became apparent that these groups had in fact identified the same protein: ‘PHF protein’ was tau, in a hyperphosphorylated state.

A key discovery came in 1979 with the recognition that the PHF protein originated from neuronal microtubules. Published in the Lancet, the New York group, led by Grundke-Iqbal, demonstrated immunological cross reactivity between rabbit polyclonal antibodies raised against normal human neuronal microtubules, and the PHF protein [16]. The paper presciently concluded ‘whatever this antigen may be, a role for neurotubules is indicated in the biochemical origin of neurofibrillary tangles’.

The missing link was made in 1986, when the same group demonstrated that antibodies to the PHF protein reacted specifically with tau [17], that antibodies to tau recognized some NFTs in post-mortem AD brain samples, and that this immunolabelling significantly increased after the tissue sections were chemically dephosphorylated [18]. These three pieces of information confirmed that the PHF protein identified in 1974 was in fact tau in a normally phosphorylated state, and that NFTs comprised abnormally phosphorylated tau.

### 2.3. Tau May Be Detected in the CSF as A Biomarker of AD

A 1985 Lancet publication detailed the first successful detection PHF protein antigens in the CSF of patients with AD, using a competitive-inhibition ELISA technique [19]. Despite considerable overlap between the groups, when compared to 9 control patients with non-AD neurological diseases (including stroke, epilepsy, and multiple sclerosis), the 9 AD patients had a significantly greater CSF PHF antigen level. The implication of this finding—that a biomarker-based diagnostic test for AD was feasible - was not explored in the brief half-page report, but was made explicit in a replication study published 3 years later: ‘Cerebrospinal Fluid-Based Laboratory Test for Alzheimer’s Disease’. Using the same ELISA technique, elevated CSF PHF protein levels were identified in AD patients across 3 separate analyses, however interpretation was similarly stymied by overlap between groups [20]. The authors concluded that an elevated CSF PHF concentration ‘may serve as a useful adjunct to the diagnosis and therapeutic response when combined with appropriate clinical neuropsychological findings’ nevertheless pessimistically conceded that ‘because of the complex nature of the pathology, it is possible that no single biochemical marker will be specific for the diagnosis of AD’.

Meanwhile, in 1987, an alternate technique was published using Western blot CSF analysis to detect the presence of Alzheimer-related neuronal protein A68 (now known to represent a triplet of phosphorylated tau: tau55, tau64, and tau69, also called PHF-tau [21]). When concentrated, purified CSF was analyzed, A68 was present in the CSF of patients with AD (n = 9), but not in CSF samples from non-demented patients (n = 6) [22]. The authors concluded that the detection of A68 in CSF could reliably distinguish between AD and non-AD patients, unlike the previous PFH protein ELISA techniques. Relatively large volumes of CSF were required for Western blot, a barrier the authors suggest could be overcome by development of an ELISA-based technique using A68.

The optimal combination of a highly sensitive, technically feasible test for CSF total tau (t-tau) was finally established by Vandermeeren in a 1993 paper showcasing an ELISA using a monoclonal capturing antibody directed at tau (AT120) and rabbit anti-human tau antiserum [23]. This assay had a tau detection limit less than 5 pg/mL in CSF. CSF samples from 190 patients were assayed including patients with AD (n = 27), non-AD neurological diseases (n = 129) and controls (n = 51). AD patients had CSF t-tau levels approximately 100 times greater than control patients (CSF t-tau AD 10.9 ± 4.9 pg/mL vs. control 0.1 +/− 0.5 pg/mL).

In the wake of this ground-breaking paper, 1995 saw the simultaneous publication of 3 further ELISA techniques to measure CSF t-tau [2,24,25]. Along with Vandermeeren’s original technique, with slight variations these remain the 4 methods still used to measure CSF t-tau today (see Table 1). Each uses different capturing and monoclonal detection antibodies directed at various tau epitopes to recognize all tau isoforms regardless of phosphorylation status. All these seminal publications reported that their assay detected significantly elevated CSF t-tau levels in AD patients compared to controls. Absolute values vary widely between studies, likely due to technical differences in the assays used and differences in cohort demographics. The latter is particularly true when considering the remarkable relative increase between cohorts in Vandermeeren’s publication: the control group included participants as young as 5 (maximum age 80; mean age 44) compared to the AD group’s mean age of 67 (range 47–88).

## 3. Tau: The Molecule

### 3.1. Tau’s Expression and Structure

Tau is abundantly expressed by neurons in the central nervous system (CNS), mostly localizing to the microtubule-rich axon to facilitate intraneuronal transport and maintain neuronal structural integrity. Smaller quantities of tau in the somatodendritic compartment modulate post-synaptic receptor activity [26]. Despite its initial characterization as a neuron-specific protein [27], transcriptome studies have revealed that some glial cells including oligodendrocytes and astrocytes also express low levels of tau [3]. Beyond the CNS, tau is expressed in peripheral nervous system (PNS) axons in a higher molecular weight form, appropriately dubbed ‘big tau’ [28].

Tau is encoded by a single gene, MAPT (microtubule associated protein tau), located on the long arm of chromosome 17 at 17q21.3. Six tau isoforms exist in the adult CNS, produced via alternative splicing of exons 2, 3 and 10 from this gene [29]. The bigger PNS isoform owes its higher molecular weight to the inclusion of exon 4a [30]. In the CNS, the differing exon content results in either 3 or 4 microtubule binding repeats comprising 31 to 32 amino acids each in the isoform’s carboxy terminal half (referred to as 3R taus and 4R taus, respectively), and 0, 1, or 2 terminal inserts comprising 29 amino acids each (0N, 1N and 2N, respectively). The various possible arrangements result in the expression of the 6 isoforms: 3 3R taus (0N3R, 1N3R, 2N3R) and 3 4R taus (0N4R, 1N4R, 2N4R) (see Figure 1). The largest isoform, 2N4R tau, is 411 amino acids long; the smallest isoform, 0N3R tau, is 352 amino acids long. Adult brains express 3R and 4R taus in a one-to-one ratio [31]. Terminal insert number is less evenly distributed: 1N, 0N and 2N isoforms account for 54%, 37% and 9%, respectively, of total brain tau [29]. Alternate splicing is developmentally regulated such that fetal brains only express the smallest tau isoform (0N3R), while all 6 isoforms are present in the adult brain.

In the tauopathy brain, it is hypothesized that a disruption to the physiological equimolar 3R:4R tau ratio may contribute to tau’s toxicity. For example, some familial tauopathies are associated with splicing mutations that result in a surplus of 4R tau isoforms [32], while disrupted 3R:4R ratios have been demonstrated in NFT-bearing neurons in the AD brain [33]. This suggests that neuronal viability relies on accurate regulation of tau alternative splicing, and maintenance of the physiological 3R:4R tau isoform balance.

Tau is structurally subdivided into four regions: the N-terminal acidic region (containing 0–2 inserts), a proline-rich domain, the microtubule binding domain (MTBD; containing 3–4 microtubule binding repeats) and the C-terminal region. Depending on the tau isoform, epitopes vary across these regions [34,35].

Alternative splicing influences tau’s interactions with tubulin. Isoforms with a second microtubule binding repeat (i.e., 4R taus) and more amino terminal inserts demonstrate enhanced tubulin binding, making 2N4R tau the most effective and 0N3R tau the least effective isoforms when it comes to microtubule assembly [36]. The enhanced binding of 4N taus is attributed to the inter-repeat sequence between the first and second microtubule binding repeat, which is absent in 3N taus, and happens to have more than double the binding affinity of any individual binding repeat [29].

### 3.2. Tau’s Function

Tau’s vital role in neuron physiology is evidenced by its phylogenetic conservation. Besides primates and other mammals, tau-like homologues have been identified in nematodes [37], bullfrogs [38], Drosophila [39], and goldfish [40].

Tau was originally named and defined by its ability to stabilize microtubules. Microtubules form a molecular scaffold upon which neuronal structural integrity depends. Each microtubule comprises tubulin dimers arranged into a hollow polymer tube of variable length. More than mere physical struts, microtubule arrays also serve as molecular conveyor belts to transport substances such as organelles within the neuron [41]. These axoplasmic transport networks are essential to neuronal function, particularly considering that the distance between dendrite and axon terminal in humans can exceed one meter.

Microtubules are dynamic structures that vacillate between phases of gradual growth and rapid truncation to meet the cell’s needs [42]. This length alternation, known as dynamic instability, relies on the highly regulated addition or subtraction of tubulin subunits to the microtubule polymer [43]. Choreographing these changes, while maintaining microtubule stability, depends on a family of proteins called microtubule associated proteins (MAPs). Tau is the primary neuronal MAP; other minor MAPs include MAP1, MAP2 and MAP4 [44].

Tau also acts as a MAP in oligodendrocytes, the cells responsible for myelinating CNS axons. Oligodendrocytes express tau in their membranous cellular processes where it stabilizes microtubule networks during process formation, axonal contact, and myelination [45]. In astrocytes, the trace levels of tau do not appear to serve a major role, while it is unclear if microglia normally contain tau or if it only accumulates under pathological conditions [45]. Increasing evidence suggests that microglia are implicated in tau propagation within the brain [46].

Beyond promoting microtubule assembly and stabilization, tau’s diverse roles include regulating axonal transport, orchestrating axonal elongation and maturation, facilitating the formation of actin filaments, interacting with membrane proteins, and modulating NMDA receptor signaling (see Figure 2) [4,47]. Tau is also present in the nucleus under physiological conditions. Acute oxidative stress has been demonstrated to drive tau to the nucleus, suggesting that it may be important for preserving DNA integrity [48]. Nuclear tau also induces chromatin relaxation, with downstream implications for global transcription alterations [49].

## 4. Conversion to a Pathological Tau State

The sequence of events required to shift tau from a physiological to a pathological state remain unclear. The process of tau aggregation requires a highly soluble protein with an ill-defined secondary structure to be incorporated into highly regular, insoluble polymer. Clearly, significant structural modifications must occur to facilitate this transition.

The tau protein is subjected to an array of post-translational modifications including phosphorylation, acetylation, methylation, ubiquitylation, glycation, glycosylation, nitration and prolyl-isomerization [47]. Each modification has implications for tau’s metabolism, aggregation, and function, including microtubule binding. Modifications occur under both physiological and pathological conditions, with some modifications being seen in both states.

Extensive analyses of PHF cores reveal tau in both a truncated [50,51,52,53] and hyperphosphorylated [18] state. Consequently, both of these post-translational modifications have been mooted as major molecular events required for tau polymerization and PHF formation to occur. It is likely that both events are important in the formation of pathological tau aggregates. With increasing phosphorylation, tau’s affinity for microtubules dwindles, eventually leading to tau’s detachment from the microtubule altogether. Since truncation promotes and accelerates tau aggregation, post-translational truncation of this unbound hyperphosphorylated tau would be expected to facilitate its aberrant accumulation.

It is increasingly recognized that tau’s neurotoxicity is driven in large part not by insoluble NFTs, but by pre-fibrillar tau oligomers formed early in the aggregation process [54]. These small, soluble species comprise monomeric or dimeric subunits of hyperphosphorylated or pathologically truncated tau. Oligomeric tau has been demonstrated to impede myriad neuronal processes including genomic regulation, protein degradation, energy metabolism, intraneuronal transport and synaptic signaling [55,56]. The mechanisms by which oligomeric tau elicits neurodegeneration remain unclear, though may be related to the species’ propensity to diffuse into the extracellular space and propagate between neurons [56].

### 4.1. Truncation

Proteolysis may confer tau a toxic gain-of-function, generating truncated species capable of instigating and perpetuating abnormal accumulation. In transgenic rat models, truncated tau can trigger and drive neurofibrillary degeneration, including sequestering endogenous rat tau into pathological complexes [57,58]. In the human AD brain, a large proportion of NFTs contain tau truncated at either or both of its N-terminal or C-terminal ends [59]. Tau C-terminally truncated at Glu-391 is a major constituent of the PHF core [51,52]. Using mAb423, a monoclonal antibody which specifically recognizes this truncation, Glu-391 truncated tau has been associated with early and advanced neurofibrillary pathology in the AD brain [60,61].

In vitro studies have shown that tau’s microtubule binding domain is essential for polymerization; conversely, the C-terminus inhibits tau aggregation. C-terminus deletions as small as 12 amino acids and as large as 121 amino acids (respectively: deleting amino acids 430-441 inclusive and 321-441 inclusive) significantly potentiate the rate and extent of tau polymerization [62]. The ‘aggregation protection’ offered by the C-terminus is demonstrated by its ability to inhibit polymerization when added in synthetic form to full-length or truncated tau preparations [63]. Regarding the minimum tau length required for polymerization, while 321-441 truncated tau can still polymerize (despite this truncation including microtubule binding repeat [MTBR] 4 and most of MTBR 3), extending the truncation just 7 amino acids further into MTBR 3 (deleting amino acids 314-441 inclusive) yields a peptide that fails to form filaments [62]. This suggests that only part of tau’s microtubule binding domain (repeats 1, 2 and part of 3) is required for tau to polymerize.

### 4.2. Phosphorylation

Phosphorylation is tau’s major post-translational modification in terms of both the number of sites along the molecule at which it may occur [34], and historical research attention. The latter can perhaps be explained by the early realization of the importance of phosphorylation in NFT pathology. In 1986, the New York team including Iqbal and Grundke-Iqbal published 3 papers in quick succession. May’s publication confirmed that PHFs comprised tau [17], a July publication identified tau’s hyperphosphorylated state within the NFT, and a third Lancet publication in August identified that it is tau’s abnormal phosphorylation per se that inhibits microtubule assembly [64]. In this final paper, fresh post-mortem brains from AD and non-AD controls were examined. As expected, tau was abnormally phosphorylated in AD brains, but not in controls, and in vitro microtubule assembly was only observed in non-AD brains. When AD brain extracts were added to normal brain samples, microtubule assembly proceeded normally, arguing against the presence of a microtubule assembly inhibitor in the AD brain. Crucially, microtubule assembly could be induced in the AD brain by the addition of an artificial tau mimic (the polycation DEAE-dextran, a microtubule assembly stimulant). These findings suggested that in AD, rather than a tubulin defect or the presence of an inhibitor, defective microtubule assembly was due to tau being rendered dysfunctional by its hyperphosphorylated state.

Phosphate groups may be added to the tau molecule at one of three amino acids: serine (Ser), threonine (Thr) and tyrosine (Tyr). Considering the largest tau isoform, 2N4R tau, phosphorylation can occur at 85 potential sites. Most of these sites are serines (53%) or threonines (41%) rather than tyrosines (6%) [35]. Normal tau contains 2 to 3 moles of phosphate per mole of tau, a ratio that optimizes tau’s interaction with tubulin and its promotion of microtubule assembly [65]. Cytosolic hyperphosphorylated tau from AD brains has double or triple this phosphate to tau molar ratio: 5 to 9 moles of phosphate per mole of tau [65].

Tau hyperphosphorylation leads to multiple pathological downstream events. The resulting molecule has reduced affinity for microtubules and is more likely to aggregate and form fibrils [66]. Healthy neurons demonstrate a spatial distribution of tau such that axonal concentrations are greater than those in the somatodendritic compartment [67]. In tauopathies this gradient is flipped, with mislocalization of tau to the somatodendritic compartment being a prominent feature [35]. Accumulated in the dendritic spines, hyperphosphorylated tau disrupts synaptic function via several mechanisms, including synaptic anchoring and impaired trafficking of glutamate receptors [68], as well as promoting synaptic excitotoxicity via interactions with Aβ oligomers [69]. Aberrant tau relocation is thought to occur early in the pathophysiology of tauopathies and may account for clinically apparent brain dysfunction in the absence of evidence of neurodegeneration [68]. Additionally, hyperphosphorylated tau is heavily implicated as a driver of Aβ-induced cell death [70].

It is unclear why tau becomes hyperphosphorylated in the first place, although evidence suggests that it may be related to aberrant kinase or phosphatase activity [71,72]. Indeed, tau’s degree of phosphorylation is a consequence of the action of protein kinases, which promote phosphorylation, and phosphatases, which remove phosphate groups. The former enzymes fall into three classes: proline-directed serine/threonine protein kinases (e.g., MPAK, Cdk5, GSK-3), non-proline-directed serine/threonine protein kinases (e.g., TTBK1/2, CK1, DYRK1A, MARK, Akt, PKA, PKC, AMPK and CaMKII), and tyrosine kinases (e.g., Fyn, Src, Abl, Syk) [34]. Dephosphorylation occurs via a range of phosphatases such as PP1, PP5, PP2B and PP2A, with PP2A accounting for over 70% of the phosphatase activity in the brain [73].

Not all phosphorylation is pathological: short-term increases in tau phosphorylation, such as those that occur during sleep and in early brain development, are thought to be an adaptive process to promote neuronal plasticity. Tau phosphorylation follows a circadian rhythm whereby sleep-associated hypothermia triggers increased phosphorylation, possibly due to PP2A inhibition [74]. Hibernating animals exhibit similar hypothermia-driven tau hyperphosphorylation [75]. Sustained hyperphosphorylation is characteristic of the embryonic and early developing human brain, where tau phosphorylation levels are on par with those seen in AD brains [76]. With maturation, phosphatase activation lowers tau phosphorylation levels to those seen in the normal adult brain [77]. Both growing infant and sleeping adult brains are in a state of heightened neuroplasticity. Physiological tau hyperphosphorylation renders neuronal cytoskeletons less stable and more flexible: an ideal conformation in such a dynamic state. In the developing brain, cytoskeletal flexibility is further enhanced by restricting tau expression to one isoform, N0R3 tau, which has the lowest microtubule affinity of all the isoforms.

## 5. Tau and Neurodegeneration

Tauopathies describe a phenotypically diverse range of neurodegenerative diseases united by the same underlying brain pathology: intracellular aggregations of hyperphosphorylated tau (see Figure 3). Wide variation is seen in the tau lesional morphology and isoform content, cell types involved, and clinical presentation (see Table 2). AD is perhaps the best known tauopathy, however over 25 familial and sporadic tauopathies have been identified. 

The term ‘tauopathy’ was coined in 1997 by Spillantini in describing a newly identified disorder: ‘multiple system tauopathy with presenile dementia’ (MSTD) [78]. Clinically, this autosomal dominant disorder was characterized by dementia, disinhibition, bradykinesia, rigidity, and superior gaze palsy, and pathologically by abundant, widespread hyperphosphorylated fibrillary tau deposits without Aβ pathology. MSTD was soon understood to belong to a group of familial tauopathies associated with MAPT gene mutations, collectively referred to as frontotemporal dementia with parkinsonism linked to chromosome 17 (FTDP-17).

With increasing genetic understanding of tauopathies, FTDP-17 is now considered a familial form of frontotemporal lobar degeneration due to tauopathy (FTLD-tau) [79]. The ‘-tau’ suffix distinguishes these forms of FTLD pathologically from those due to deposition of one of two other proteins: the TAR DNA binding protein of 43 kDa (TDP-43), and the fused in sarcoma protein (FUS), referred to as FTLD-TDP43 and FTLD-FUS, respectively. Clinically, FTLD has three main clinical phenotypes. The behavior variant features disinhibition, apathy, and prominent personality changes. Those presenting with predominantly language symptoms, known as primary progressive aphasia, may have either a non-fluent (halting, effortful speech) or semantic (word-finding difficulties) presentation.

FTLD-tau disorders are primary tauopathies: those where tau deposition is the predominant feature. Secondary tauopathies describe disorders that require additional etiologies such as Aβ deposition (i.e., AD), repeated head trauma (i.e., CTE), or autoimmunity (e.g., anti-IgLON5-related tauopathy) for tau deposition. Tauopathies may be further distinguished by the predominant isoform content of the tau inclusions: 4R tau (e.g., progressive supranuclear palsy, corticobasal degeneration, globular glial tauopathy), 3R tau aggregations (e.g., Pick disease), or mixed (e.g., AD) (see Figure 4).

## 6. Tau as a Biomarker in Biological Fluids

Tau’s role as diagnostic, prognostic and treatment-response biomarker has been the focus of intense research interest over the past four decades. Tau’s development as a biomarker is inextricably tied to advances in our pathophysiological understanding AD, the most prevalent and well-studied tauopathy.

The inclusion of biomarker parameters in the latest research diagnostic criteria for AD [5] represents a shift in definition from a syndrome-based to a biological framework [6]. Traditionally, diagnosis has relied on the application of clinical criteria outlining typical symptoms and signs. Due to overlapping features with other neurodegenerative disorders, more accurate, sensitive, and reliable diagnostic tools were thus in high demand. Breakthroughs in the AD research space, particularly the molecular characterization of the NFT in the 1980s, made the search for antemortem biomarkers of AD feasible. The CSF, in direct communication with the brain’s extracellular space and thus likely to reflect pathological alterations therein, was the natural substance in which to begin the search for these biomarkers. With the development of more sensitive detection methods, the blood–into which CSF biomarkers of neurodegeneration may be absorbed from the subarachnoid space–offered a second more readily sampled fluid to test for biomarkers.

### 6.1. Caveats and Challenges in the Interpretation of Tau as a Biomarker

Tau’s abundance in the CSF reflects neuronal damage of any etiology. If this neurodegeneration is not due to an underlying tauopathy, the liberated tau will be in a non-pathological state (i.e., non-phosphorylated and untruncated). For example, CSF t-tau peaks after an ischemic stroke, while CSF p-tau remains normal [80]; the same pattern is seen in the setting of mild head trauma among amateur boxers [81]. Conversely, in the setting of tauopathy-driven neurodegeneration, the tau released also includes species which are in a pathological state (i.e., phosphorylated and truncated).

The modern model of AD, particularly when it comes to research frameworks, is divided syndromically into three cognitive stages occurring on a continuum [6], reflective of progressive underlying pathological change. The preclinical stage of AD denotes cognitively unimpaired (CU) patients in whom the pathological process of AD has been initiated. In the brains of such patients, abnormal Aβ peptide processing is leading to the deposition of Aβ plaques in the brain. The key biomarker of this phase is a reduction in CSF Aβ42 concentrations, reflecting its sequestration into plaques. After a variable lag period comes the prodromal stage, during which the patient is said to have AD with mild cognitive impairment (MCI). As the name suggests, subtle cognitive impairments typically affecting short term memory become clinically apparent, reflecting the onset of underlying neurodegeneration. Ongoing neuronal loss sees progressive cognitive impairment across multiple domains with associated functional impairment, denoting the transition to the third and final stage: AD dementia.

This model partly explains why p-tau elevations in AD correlate with cognitive decline better than Aβ42 reductions, which reflect early or even preclinical pathology [82]. A longitudinal study following the conversion of patients from prodromal AD to dementia found that while all patients had fully suppressed CSF Aβ42 levels at baseline, rapid converters (those who converted to dementia within 5 years) could be distinguished from slow converters (between 5 and 10 years) by a baseline elevation in p- and t-tau [83]. This implies that Aβ42’ s plateau denotes an earlier neuropathological stage—at least 5 to 10 years prior to conversion to dementia–than tau biomarker elevations, which herald more imminent clinical decline. While AD may clinically progress along a linear continuum that reflects progressive neurodegeneration, the accumulation of biomarker abnormalities may not occur in parallel. For example, CU patients may have positive p-tau biomarkers for years before MCI becomes apparent. In fact, the biomarker definition of AD across all three cognitive stages is the same: abnormalities in both Aβ42 and p-tau, with or without t-tau elevation. The corollary is that the accumulation of biomarker abnormalities is not absolutely tied to clinical progression: a patient with this AD biomarker profile may syndromically be in any of the three cognitive stages, and this is likely determined by a variety of other factors including comorbid brain pathology, cognitive reserve, and other factors.

A key and as-yet unresolved issue is why CSF p-tau concentrations are not elevated in other tauopathies besides AD, given that they all also feature neurodegeneration and aggregation of hyperphosphorylated tau. Multiple possible explanations have been put forward to explain this unusual phenomenon. It may be that existing CSF p-tau assays do not detect the epitopes that are phosphorylated in non-AD tauopathies [84]. However, this explanation would not account for the fact that some tauopathies, particularly progressive supranuclear palsy, may demonstrate significantly decreased CSF p-tau concentrations even compared to controls [85]. It is more likely that the absence of CSF p-tau elevations in non-AD tauopathies reflects important structural and functional differences in the pathological tau isoforms seen across the tauopathy continuum. Different tau isoforms demonstrate different kinetics [86]. For example, 4R taus are more rapidly metabolized than 3R isoforms, phosphorylated tau species have shorter half-lives than their corresponding non-phosphorylated species, and isoforms differ in the rate at which they are released from the neuron [86], with neurons that overexpress MAPT mutations releasing significantly less extracellular tau [87]. Even in the absence of mutations, posttranslational modifications may result in tau species that are undetectable by current assays. For instance, even when isoforms are identical, as in the case of the 4R tauopathies PSP and CBD, differences in proteolytic cleavage result in the pathological species having different C-terminal amino acids, with likely implications for the kinetics of these molecules [88]. 

In patients with AD, CSF t-tau and p-tau elevations may precede clinical symptom onset by decades. The aforementioned model of AD [6] proposes that the appearance of these soluble tau species in the CSF is a reflection of neuronal death with associated passive release of previously intracellular tau in both normal (t-tau) and hyperphosphorylated (p-tau) forms. However, CSF t-tau and p-tau concentrations in AD patients only modestly correlate with NFT pathology measured by tau-PET [89]. While some autopsy studies have demonstrated a correlation between CSF p-tau181 and the presence of AD tangle pathology in cortical biopsies [90,91], others were unable to identify any correlation [92]. Furthermore, the rate of increase in p-tau181 concentrations has been found to decrease with increasing neurodegeneration [93]. These observations suggest that there are important differences in the soluble and aggregated tau species in the pathophysiology AD, and that cerebral Aβ may trigger the unique tauopathy of AD [94]. Specifically, Aβ may actively drive the secretion of tau from the neuron, accounting for the pathological CSF tau concentrations seen early in AD pathogenesis, before frank neurodegeneration [95].

Tau’s ability to be secreted into the CSF, rather than appearing there only as a result of neurodegeneration, is supported by the observation that low levels of CSF tau are present in healthy control patients. Values up to 300 pg/mL are generally considered normal [96], however mean values up to 375 pg/mL [97] have been reported in some control populations. Rather than representing chronic low-level neurodegeneration, evidence is mounting that tau is not exclusively an intracellular protein: neurons may actively secrete tau under certain physiological conditions. Secretion stimuli and mechanisms remain poorly understood, however in vitro studies have demonstrated increased neuronal activity as a secretion trigger [98], with tau entering the extracellular space via vesicular and alternative unconventional secretory pathways [87,99]. Importantly for the diagnostician, this constitutive tau secretion generates CSF tau levels that are orders of magnitude lower than those seen in AD. Nevertheless, the existence of pathways for tau to emerge in the CSF in the absence of neurodegeneration lends support to the theory of pathological Aβ-driven tau secretion early in AD.

### 6.2. Tau in the CSF

ELISAs have been developed to analyze the CSF for concentrations of both t-tau, encompassing both phosphorylated and non-phosphorylated forms, and phosphorylated tau (p-tau). Assays for the latter may detect tau phosphorylated at epitopes known to occur in disease states, such as threonine-181 (p-tau181), threonine 231 (p-tau231) and serine-199 (p-tau199) [100]. More recently, assays targeting tau phosphorylated at other epitopes, such as threonine-217 (p-tau217), have become available [101]. As the number of novel CSF tau biomarker assays increases, the role for each in the diagnosis, prognosis, and management of patient with tauopathies is rapidly being delineated.

#### 6.2.1. CSF T-Tau

CSF t-tau concentrations are thought to reflect the extent and intensity of neuronal damage occurring in the brain. For example, the magnitude of CSF t-tau increase following an ischemic stroke correlates with the volume of tissue damage calculated using neuroimaging [102]. CSF t-tau levels measured 2–3 days after a severe traumatic brain injury predict clinical outcomes at 1 year, with higher levels predicting worse outcomes, likely due to a greater extent of axonal damage [103]. Even minor axonal damage is reflected in CSF t-tau levels, with elevated concentrations detected in amateur pugilists after a boxing bout compared to levels following an extended absence from the ring [81]. Decades after retirement, former athletes with a history of multiple concussions have been found to have elevated CSF t-tau concentrations, with demonstrable axonal damage reflected by reduced white matter tract integrity on diffusion tensor imaging [96].

Even before tau was recognized as the key protein comprising NFTs, experiments in the 1980s had identified the increased concentration of ‘PHF antigen’ in the CSF of patients with AD compared to controls [19,20]. Multiple studies have since replicated this finding, with a recent meta-analysis of 151 studies (AD n = 11,341; control n = 7086) reporting an average AD to control ratio of 2.54 for CSF t-tau concentration, with all included studies having a ratio above one [104].

CSF t-tau demonstrates utility as a marker of AD severity. In a longitudinal study of 142 patients with AD, those with higher CSF t-tau concentrations (>800 ng/L, n = 35) had a lower MMSE score at baseline and experienced more rapid reduction in this score over 3 years than those with lower CSF t-tau concentrations (<800 ng/L, n = 107) [105]. Higher CSF t-tau levels are seen among non-amnestic compared to amnestic AD patients, thought to reflect the greater extent of cortical involvement in the former group [106].

There is a potential disease monitoring role for CSF t-tau in AD, with increasing concentrations correlating with degree of cognitive impairment. A significant progressive increase in average CSF t-tau concentrations has been found when comparing patients with early AD (269 ng/L) to those with more advanced disease stages (468 ng/L and 495 ng/L among mild and moderate AD patients, respectively) [82].

Besides AD, stroke, and traumatic brain injuries, elevated t-tau concentrations may appear in the CSF of patients with sporadic Creutzfeldt-Jakob disease (sCJD), dementia with Lewy bodies (DLB), and some forms of frontotemporal dementia (FTD) [107]. Average CSF t-tau levels for DLB and FTD fall between those seen in controls and patients with AD, while sCJD has the dubious distinction of the highest recorded CSF t-tau levels [107,108].

#### 6.2.2. CSF p-Tau

Following 1986’s trio of publications highlighting the role of tau phosphorylation in NFT pathology, the development of ELISA techniques to detect CSF p-tau began in the mid 1990s. The first published assay was described in 1995, in the same paper from Blennow detailing his novel CSF t-tau assay [2]. Using AT180 and AT270 as capturing antibodies, and HT7 and AT120 as detection antibodies, concentrations of tau phosphorylated at threonine 181 (p-tau181) and threonine 231 (p-tau231) were found to be increased approximately 3.5-fold in the CSF of AD patients compared to controls (AD n = 44; average p-tau concentration 2230 ± 930 pg/mL vs. controls n = 31 average p-tau concentration 640 ± 320 pg/mL).

A multitude of subsequent studies have demonstrated that CSF p-tau concentrations are consistently elevated in patients with AD compared to controls. A meta-analysis of 89 such studies reporting CSF p-tau concentrations using single or multiple phosphorylation sites identified an AD to control ratio of 1.88 (AD n = 7498; control n = 5126) [104], with all 89 studies reporting a ratio greater than one. Importantly, CSF p-tau concentrations in AD are not only elevated compared to controls, but also to patients with other tauopathies and neurodegenerative disorders [109]. The unique degree of CSF p-tau elevation in AD is of great utility in distinguishing AD from clinically similar differential diagnoses. Lower elevations in CSF p-tau may also be seen in DLB and CJD [107], and a small proportion of patients with Parkinson’s disease dementia [110].

CSF p-tau181 is the most thoroughly examined p-tau species as a biomarker of AD, included in most assays used in modern AD diagnostics. Nevertheless, emerging evidence suggests that the concentrations of other p-tau species may provide equivalent, additional, or superior diagnostic information compared to p-tau181. A shortage of epitope options to assay is not a barrier: analyses of PHFs from AD brains using antibody or mass spectrometry techniques have identified over 60 phosphorylation sites [111]. 

Assays for the concentration of p-tau231 and tau phosphorylated as serine-199 (p-tau199) tightly correlate with p-tau181, with all three epitopes found to be equally and significantly raised in AD patients compared to patients with other dementias, and controls [112]. In the same study, diagnostic accuracy varied only slightly when applying ideal values defined in a consensus report for diagnostic biomarkers in AD [113]. Assays for all three p-tau epitopes met an 85% specificity minimum when differentiating AD from non-AD, however a 75% sensitivity minimum was only met by p-tau181 and p-tau 231. CSF p-tau231 has not consistently demonstrated such robust specificity: in another study, levels did not correlate with dementia severity in AD and could only discriminate between AD and non-AD dementias, including FTD, vascular dementia, and DLB with a specificity of 80% [114]. Instead, the key role for CSF p-tau231 appears to be as a biomarker of very early AD. Detecting pre-clinical AD as early as possible is crucial to the success of therapeutic interventions in AD. To date, CSF p-tau231 is the earliest identified p-tau species to become elevated in AD. A 2022 Lancet publication identified that CSF p-tau231 was sensitive to Aβ deposition in brain regions typically affected early in AD, such as the medial orbitofrontal, posterior cingulate and precuneus, before Aβ PET positivity [115].

Recent data from two independent studies suggests that CSF p-tau217 may outperform CSF p-tau181 as a diagnostic biomarker of AD. In the first study, CSF p-tau217 concentrations among prodromal and AD dementia patients were several-fold higher than CSF p-tau181 in the same patients, and demonstrated greater annualized increases longitudinally [101]. CSF p-tau217 also outperformed p-tau181 as a predictor of tau PET abnormalities and correlated more strongly with CSF and PET measures of Aβ burden. Further, p-tau217 demonstrated superior diagnostic accuracy in distinguishing AD dementia from non-AD neurodegenerative disorders. Furthermore, CSF p-tau217 also outperformed p-tau181 when separating CU individuals with positive Aβ scans from those with negative scans, and when distinguishing the former group from those with AD dementia. The second study reinforced these findings, reporting a six-fold increase in CSF p-tau217 concentrations among AD patients compared to p-tau181’s mere 1.3-fold elevation, relative to controls [116]. These findings were validated in a second cohort of patients without cognitive impairment or with MCI, among whom CSF p-tau217 outperformed p-tau181 as a predictor of Aβ PET positivity. Overall, these findings suggest that p-tau217 is a better biomarker of AD than p-tau181 in both the pre-clinical and advanced stages of AD. Further, the correlations between CSF p-tau217 and Aβ PET data suggest a link between the pathophysiology of these proteins, with potential therapeutic applications.

#### 6.2.3. CSF Tau Profiles: Diagnostic Utility, Ratios, Disease-Specific Profiles, and Longitudinal Changes

Interpreted in combination, CSF t-tau and p-tau levels may provide clinically useful diagnostic, prognostic, and treatment-response information among patients with neurodegenerative disorders, although their widespread utilization has been limited by cost, access to analytical technology and invasiveness of lumbar puncture.

Specific patterns of CSF t-tau and p-tau concentrations, along with Aβ42 levels, have been characterized in various dementia diagnoses. In a large cohort of dementia patients, the combined interpretation of CSF p-tau and Aβ42 concentrations correctly classified 92% of AD patients and 88% of controls [107].

An elevated CSF t-tau but a low CSF p-tau (i.e., a high CSF t-tau/p-tau ratio) is a specific indicator of the presence of neurodegeneration in the absence of an underlying tauopathy. sCJD is the sine qua non here, where the combination of elevated CSF t-tau and a high t-tau/p-tau ratio, using minimum cut-offs of 1400 pg/mL and >25, respectively, has a specificity of 99% [117].

Among patients with MCI, higher CSF t-tau and p-tau levels at diagnosis have consistently been associated with a higher risk of evolution to AD [118,119,120,121]. Whether the degree of elevation correlates with the rate of transition, however, varies across studies. For example, of 14 MCI patients who developed AD during a follow-up period of 3–12 years, the 50% of patients with the highest CSF t-tau and p-tau at baseline progressed to AD within a shorter timeframe than those with levels in the lowest 50% [122]. Conversely, this relationship was not observed among a larger cohort of 57 MCI patients who went on to develop AD during a mean follow up time of 4.0–6.8 years [118].

CSF tau profiles have shown promise in dividing AD patients into clinically meaningful phenotypes. A cross-sectional study that stratified AD patients (n = 177) into three subsets based on low, high or very high CSF t-tau and p-tau levels found that the ‘very high’ group (n = 17, mean CSF t-tau 1720 ± 430 pg/mL, mean CSF p-tau 170 ± 38 pg/mL) had a distinct cognitive profile characterized by worse performance on tests of memory, mental speed and executive functions [123]. These differences could not be accounted for by differences in disease duration, severity (as measured by MMSE) or functional impairment, suggesting that this CSF tau pattern denotes a specific cognitive profile among patients with AD. A subsequent prospective study using the same 3-cluster method replicated these findings, plus further characterized the ‘very high’ profile (n = 12, mean CSF t-tau 1501 ± 292 pg/mL, mean CSF p-tau 139 ± 39 g/mL) as experiencing faster progression of cognitive deficits, higher mortality, and no response to cholinesterase inhibitor therapy [124].

Somewhat counterintuitively, at the time of AD diagnoses, patients with an elevated CSF t-tau who lack a proportional increase in p-tau have been shown to exhibit faster cognitive decline [125]. Among 151 AD patients, a low p-tau181/t-tau ratio at diagnosis was the strongest predictor of cognitive decline over time (average 2.9 points lost per year MMSE), with a dose-dependent relationship. Conversely, those patients with a CSF p-tau181/t-tau in the highest quartile, representing a proportional increase in CSF p-tau, had a more benign disease course (average annual decline of 1.3 MMSE points). This finding supports an emerging alternate hypothesis that tau phosphorylation may in fact be a protective mechanism employed by neurons to escape apoptosis in favor of chronic neurodegeneration [126]. This would explain why those AD patients with a reduced phosphorylation activity (i.e., no proportional increase in p-tau) may progress faster, owing to uncurtailed acute apoptotic neuronal cell death rather than the slower process of chronic neurodegeneration.

Interest is growing in the characterization of phosphorylation profiles for certain tauopathies. If a given tauopathy were found to be associated with the phosphorylation of a certain unique epitope or group of epitopes, highly specific assays could be developed. Further, if phosphorylation at certain epitopes occurred sequentially in a predictable fashion with disease progression, the appearance of certain phosphorylated epitopes in the CSF could permit staging of disease. 

Early progress in this tau profiling field has been made using mass spectrometry and immunoblotting techniques. Analyzing autopsied brain samples from tauopathy patients, a 2021 paper found evidence of disease-dependent phosphorylation profiles which were largely consistent between patients with the same diagnosis [127]. Certain disease-specific phosphorylation epitopes were identified: for example, Pick’s disease brains showed an excess of serine-202 phosphorylation, while phosphorylation at serine-396 was typical of argyrophilic grain dementia [127]. These early results support the idea of unique pathological mechanisms behind each tauopathy, with exciting ramifications for possible disease-specific therapeutic interventions.

Longitudinal tau profiling is an area of intense research interest. A recent study among patients with dominantly inherited AD (DIAD) captured the sequential appearance of various p-tau species across four decades of disease progression [128]. CSF concentrations of tau phosphorylated at threonine 217 (p-tau217) were elevated 21 years prior to estimated symptom onset (calculated based on the age of onset of others with the same genetic mutation). This elevation was followed by p-tau181 (19 years prior), t-tau (17 years prior), then tau phosphorylated at threonine 205 (p-tau205; 13 years prior). Near the time of symptom onset, concentrations of p-tau217 and p-tau181 significantly dropped, p-tau205’s elevation slowed, while t-tau levels continued to increase. Each of these changes was associated with other structural, metabolic, or clinical markers of disease. These findings suggest that tau phosphorylation at specific epitopes in DIAD, and potentially sporadic AD, is dynamic, and orchestrated to occur with a predictable trajectory over time. Understanding the tempo of pathological changes in tau in this granular fashion is central to the development of tau-based therapies for AD and other tauopathies.

### 6.3. Tau in the Blood

The perceived invasiveness of lumbar puncture presents a barrier to the widespread implementation of CSF tau as a biomarker beyond the research setting. Peripheral blood sampling, on the other hand, is cheaper, faster, requires less operator skill, and is more palatable for the patient. CSF reabsorption into the peripheral blood is a physiological process. Driven by a pressure gradient, CSF in the subarachnoid space enters venous blood flowing in the brain’s venous sinuses via small outpouchings called arachnoid granulations. Incorporated into the venous blood, CSF components leave the cranial vault and enter the peripheral circulation where they may be sampled using phlebotomy. 

With this basic physiology in mind, detecting CSF-derived biomarkers in the blood poses several theoretical challenges. Firstly, their concentrations will be significantly reduced. Average CSF volume is 125 mL; total blood volume is about 5000 mL. This 40-fold dilution necessitates ultra-sensitive detection methods compared to techniques used to assay CSF samples. Secondly, CSF-derived biomarkers may undergo significant, unpredictable, variable, or idiosyncratic metabolism as they pass through the liver and kidneys. Thirdly, steady state concentrations of biomarkers may vary and demonstrate different regulation between the CSF and blood. Finally, blood contains significantly more solutes than the relatively bland CSF, including clotting factors, immune system components and other plasma proteins. These components may interfere with assays or introduce other technical challenges. Furthermore, blood composition shows significantly greater variation from person to person than CSF (e.g., concentrations of hemoglobin, albumin), with this variability possibly confounding the interpretation of biomarker concentrations.

Remarkably, despite these theoretical challenges, significant progress has been made in the development of assays to measure tau as a blood-based biomarker of neurodegeneration. The evolution of these advances closely mirrors that seen in CSF tau assays: with sequential development of assays for t-tau, p-tau181, then tau species phosphorylated at other epitopes.

#### 6.3.1. Blood Total Tau (Blood T-Tau)

Early attempts to measure tau in the blood were limited by a lack of assay sensitivity. For example, in a 1995 publication assaying CSF t-tau, the authors briefly mention that ‘in a subset of patients, tau also was assayed in serum that was obtained at the same time the spinal tap was performed….notably, no tau proteins were detected in serum from any of the control subjects or patients studied here (data not shown)’ [129]. 

The low concentrations of blood-based biomarkers of neurodegeneration presented a formidable barrier for decades. Their detection required techniques an order of magnitude more sensitive that existing technology. Rather than the nanogram concentrations of CSF tau, plasma t-tau levels in MCI patients, for example, averages around 4.6 pg/mL [130], with AD levels in the realm of 37.5 pg/mL [131]. An early attempt to directly apply a CSF tau assay to human plasma failed on multiple fronts. Of 57 patients (AD n = 16, FTD n = 10, VAD n = 16, healthy controls n = 15), only 12 had detectable t-tau levels (7 of which were samples from healthy controls). No differences in plasma t-tau levels were identified between healthy controls or the dementia groups. Furthermore, plasma t-tau levels showed no correlation with CSF t-tau levels [132]. The limit of detection (LOD) of the ELISA used was not stated, however the lowest reported t-tau concentration was 200 pg/mL, suggesting that the assay’s LOD was inadequate for the intended analysis.

Inadequate LODs continued to confound results for over a decade. For instance, a 2014 study failed to demonstrate any difference in plasma t-tau or p-tau181 between AD patients and controls, however used ELISAs with LODs of 12 pg/mL for t-tau and 10 pg/mL for p-tau181 [133]. Even in studies when peripheral t-tau levels were detectable, they provided negligible clinical data. For example, following ischemic stroke, peripheral t-tau concentrations showed no correlation with infarct size, neurological deficits, or degree of disability in the early or late phase of recovery, unlike CSF t-tau concentrations [134]. Among patients with minor head injury, serum t-tau concentrations failed to discriminate between those with normal brain CTs and those with CT evidence of intracranial injury [135]. 

Given the remarkable CSF t-tau elevations characteristic of sCJD, the pre-test probability of successfully detecting peripheral t-tau, even using low-sensitivity ELISAs, was high. Indeed, in a 2011 study using an ELISA with a LOD of 18.8 pg/mL, serum t-tau was detectable and significantly elevated in all sCJD samples (n = 12), but in only 8 of the 29 samples from patients with other dementia diagnoses, and controls (AD n = 1/10 detectable; non-CJD rapidly progressive dementia n = 5/9 detectable; healthy controls n = 2/10 detectable) [136].

The development of digital ELISAs using single-molecule arrays (Simoa) heralded a new era of ultra-sensitive biomarker sampling. Additionally, known as high-sensitivity ELISAs, the technology relies on using femtoliter-sized reaction chambers capable of detecting and isolating single enzyme molecules. While existing ELISAs could detect proteins at picomolar concentrations (10^−9^ M), Simoas slashed this limit to femtomolar concentrations (10^−12^ M). A seminal paper in this field was published in 2010 by Rissin et al., reporting the successful use of Simoa to detect very low concentration of two clinically relevant serum biomarkers: prostate specific antigen (PSA) and tumor necrosis factor alpha (TNFα) [137]. Compared to existing ELISA technology, the LOD achieved using Simoa were 25-fold lower for PSA and 35-fold lower for TNFα (PSA Simoa LOD 14 fg/mL [0.4 fM] vs. existing ELISA LOD 10 fM; TNFα Simoa LOD 10 fg/mL [0.6 fM] vs. existing ELISA LOD 21 fM). 

Buoyed by this new technology, a flurry of studies reported applications of Simoa technology to blood tau measurements across a range of clinical settings. Not only could tau be detected, but it also displayed promise as a prognostic biomarker. For example, plasma tau concentrations were found to be elevated in comatose patients following cardiac arrest, with higher levels predictive of poor outcome at 6 months [138]. These early Simoa studies also contributed to our understanding of tau physiology. For instance, unique tau kinetics were identified after cardiac arrest, with serum t-tau concentrations peaking immediately after resuscitation, then some patients experiencing a second peak after 24–48 h [139]. Compared to a single t-tau peak, the presence of an additional delayed t-tau elevation was highly predictive of poor outcome at 6 months. This bimodal pattern represents two phases of neuronal injury: an acute anoxic insult followed by delayed cell death due to secondary injury, similar to the bimodal peak in troponin often seen after myocardial infarction. 

Disappointingly, in the dementia research space, early results assaying plasma t-tau with high-sensitivity immunoassays were highly varied. Confusingly, some papers reported significant reductions in plasma t-tau in AD patients compared to those with MCI and papers from one research group, both with sample sizes of less than 150, identified significantly elevated plasma t-tau among MCI and AD patients compared to controls, but could not distinguish between MCI and AD [140,141]. Another study found elevated plasma t-tau levels only in AD patients, but not in MCI patients, compared with controls [142]. The cross-sectional interpretation of plasma t-tau levels was deemed unfeasible, given that ranges significantly overlapped between diagnostic groups and failed to correlate with CSF t-tau concentrations in any group.

A study involving only MCI and control patients identified higher plasma t-tau levels among the MCI cohort (n = 161) compared to controls (n = 378), but this failed to reach statistical significance [130]. Some significant relationships were identified, however, including higher plasma t-tau levels being associated with poorer memory performance and cortical thinning in an AD-typical pattern. 

In 2016, a larger study including 1284 participants from two cohorts attempted to provide clarity in this space [143]. The Alzheimer’s Disease Neuroimaging Initiative (ADNI) cohort comprised patients with AD (n = 179), MCI (n = 195) and cognitively healthy controls (n = 189). The Biomarkers for Identifying Neurodegenerative Disorders Early and Reliably (BioFINDER) cohort contained the same categories (respectively: n = 61, n = 212, n = 274) with an additional ‘subjective cognitive decline’ group (n = 174). Like previous plasma t-tau studies, the results were mixed, with weak correlations that differed between the cohorts. For example, AD patients in the ADNI cohort had higher plasma tau compared to those with MCI or controls, but these concentrations did not correlate with CSF t-tau or CSF p-tau. Conversely, in the BioFINDER cohort, plasma tau did not vary by diagnosis, but concentrations did correlate with CSF t-tau and p-tau.

Plasma t-tau shows more promise as a prognostic biomarker, rather than a cross-sectional diagnostic biomarker. Over three years, CU patients with higher plasma t-tau levels were significantly more likely to progress to MCI, however higher levels were not predictive of conversion from the MCI to the AD dementia groups [144]. Across all groups, higher baseline plasma t-tau levels were associated with global and domain-specific cognitive decline, but results differed based on cognitive status and length of follow up. Interestingly, all findings were independent of elevated brain Aβ assessed using PET. A paper from the same group the following year identified significantly elevated t-tau and p-tau in AD plasma: p-tau181 levels were consistently associated with Aβ and tau PET, whereas plasma t-tau was associated with cortical thickness [145].

Taken together, the results of these plasma t-tau studies suggest that its role is akin to CSF t-tau’s as a non-specific biomarker of neurodegeneration. At least three observations support this hypothesis. First, changes in plasma t-tau concentrations do not correlate with Aβ or tau PET, suggesting that they are not specific to the AD pathophysiological process [144,145]. Second, plasma t-tau has been found to be elevated among those with cardiovascular risk factors known to be associated with cortical atrophy, such as diabetes, hypertension and atrial fibrillation [130]. Thirdly, higher plasma t-tau levels have been shown to correlate with reduced cortical thickness [130,145] and grey matter volume [140].

Plasma t-tau’s diagnostic utility in AD is significantly improved when it is interpreted as a ratio with plasma Aβ42. In plasma, the t-tau/Aβ42 ratio has been shown to be associated with longitudinal changes over two years in cerebral Aβ deposition, hippocampal volume change, and brain glucose metabolism [146]. This ratio also demonstrated impressive predictive power (sensitivity 80%, specificity 91%) for brain tau deposition. Of particular interest is the fact that the brain regions where this ratio strongly correlated with brain tau were classical deposition sites of NFTs in AD. These results suggest that this ratio is a powerful predictive biomarker of longitudinal neurodegeneration in AD.

#### 6.3.2. Blood p-Tau

Given CSF p-tau’s specificity for AD, research attention has shifted towards the development of high-sensitivity ELISAs for plasma p-tau. The first publication reporting plasma p-tau181 levels in AD was highly promising: AD plasma (n = 20) contained significantly greater concentrations of p-tau181 than controls [147]. Furthermore, in another cohort of 8 AD patients, plasma p-tau181 significantly correlated with CSF p-tau181. Subsequently, a number of studies have replicated these findings in larger cohorts.

Plasma p-tau181 has been shown to increase across the AD continuum [145,148,149] and demonstrate strong associations with both Aβ and tau PET [145,146,149,150], supporting its role as an AD-specific biomarker. Plasma p-tau181 is also a powerful prognostic biomarker, being predictive of one-year cognitive decline and hippocampal atrophy [149], as well as predicting at what rate this decline will occur [150]. Unlike plasma t-tau’s relationship with CSF t-tau, plasma p-tau181 shows strong, consistent correlations with CSF p-tau181 [145,151]. As a diagnostic biomarker, plasma p-tau181 can differentiate AD from frontotemporal lobar degeneration [150]. These results strongly support plasma p-tau181’s utility as a reliable in vivo biomarker of brain tau pathology.

Novel assays for plasma p-tau species phosphorylated at epitopes other than threonine-181 are rapidly being developed. Since 2020, assays to detect plasma tau phosphorylated at threonine 217 (p-tau217) and threonine 231 (p-tau231) have been the focus of large publications.

Similar to p-tau217’s flourishing reputation in the CSF biomarker realm, plasma p-tau217 is rapidly demonstrating superiority to plasma p-tau181 as a biomarker of AD. Compared to plasma p-tau181, antemortem plasma p-tau217 levels performed significantly better in differentiating AD-specific neuropathological changes from non-AD changes at autopsy [152]. In the same study, as a discriminatory diagnostic biomarker, plasma p-tau217 also outperformed plasma p-tau181 for AD versus other neurodegenerative disorders, with an accuracy on par with CSF p-tau217, CSF p-tau181, and tau PET. Plasma p-tau217 has been shown to increase early in AD and can be used to monitor disease progress. In a six-year longitudinal study, CU and MCI patients with Aβ positivity (determined by CSF analysis) had significantly elevated plasma p-tau217 compared to Aβ negative CU participants [153]. Accelerated plasma p-tau217 levels were seen in MCI patients who later converted to AD compared to stable MCI patients, and longitudinal deterioration in cognition and brain volume correlated with longitudinal plasma p-tau217 elevations.

A plasma p-tau231 assay was developed in 2021 and validated in four large independent cohorts (n = 588) [154]. With high accuracy (AUC 0.92–0.94), plasma p-tau231 could identify patients with AD, and differentiate them from Aβ-negative CU older adults. Among those with dementia, plasma p-tau231 levels could readily distinguish AD from non-AD neurodegenerative disorders and Aβ-negative MCI patients. Compared to plasma p-tau181, this novel assay identified the clinical stages of AD equally well but increased earlier in the disease process, even before Aβ PET positivity: the same advantageous characteristic typical of CSF p-tau231. Consistent with this shared characteristic, levels of plasma p-tau231 correlated strongly with CSF p-tau231. An additional strength of plasma p-tau231 is its robust predictive power for post-mortem AD neuropathology: levels taken an average of 4.2 years before death very accurately identified AD neuropathology in comparison to non-AD neurodegeneration (AUC = 0.99) in a group of patients who were all given an AD diagnosis during life.

## 7. Key Points

CSF t-tau is a general marker of neurodegeneration, elevated in a range of conditions associated with neuroaxonal injury including stroke and traumatic brain injury. Among dementia diagnoses, elevations occur in conditions including AD, DLB, and some forms of FTD.CSF p-tau elevations are consistently seen in AD.It is unclear why other tauopathies, which also feature abnormal accumulation of p-tau with neurodegeneration, do not feature elevations in tau-based biomarkers.Among AD patients, CSF p-tau181 is the most thoroughly examined p-tau species, however recent data suggest that other epitopes demonstrate superior diagnostic utility (e.g., p-tau217) or earlier elevation in the disease process (e.g., p-tau231).Like CSF t-tau, plasma t-tau elevations reflect neurodegeneration of any etiology. Higher levels are associated with cognitive decline and risk of MCI.Plasma p-tau181 levels correlate strongly with CSF p-tau181 levels. As a biomarker of AD, plasma p-tau181 demonstrates robust diagnostic and prognostic performance.Assays for other plasma p-tau species are rapidly being developed. Further research, including head-to-head comparisons to existing biomarkers, are needed to determine their role in the diagnosis and management of patients with tauopathies.

## Figures and Tables

**Figure 1 ijms-23-07307-f001:**
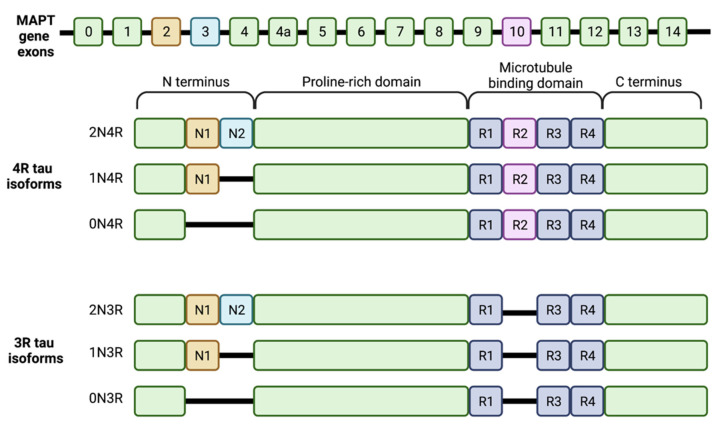
The six tau isoforms present in the human brain produced via alternative splicing of the MAPT gene’s 16 exons. Exons 2 and 3 encode the two possible N-terminal inserts N1 and N2 (shown in orange and blue). Exon 10 encodes the second microtubule binding repeat (R2, shown in pink) in the microtubule binding domain. Alternative splicing results in 6 isoforms that vary by the number of N-terminal inserts (0N, 1N or 2N) and the presence or absence of R2 (4R isoforms or 3R isoforms, respectively).

**Figure 2 ijms-23-07307-f002:**
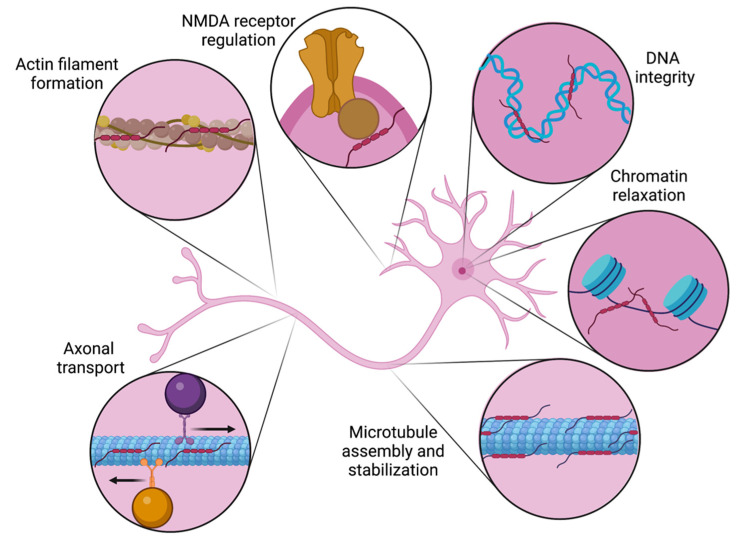
Tau’s diverse physiological roles in the neuron. Initially characterized as a protein required for microtubule assembly and stabilization, it is now recognized that tau has roles in multiple neuronal compartments. Along the axon, tau is involved regulating in bidirectional transport as well as actin filament formation. Nuclear roles include protecting DNA integrity and promoting chromatin relaxation. At the neuronal membrane, tau’s interactions with the NMDA receptor regulate its signaling.

**Figure 3 ijms-23-07307-f003:**
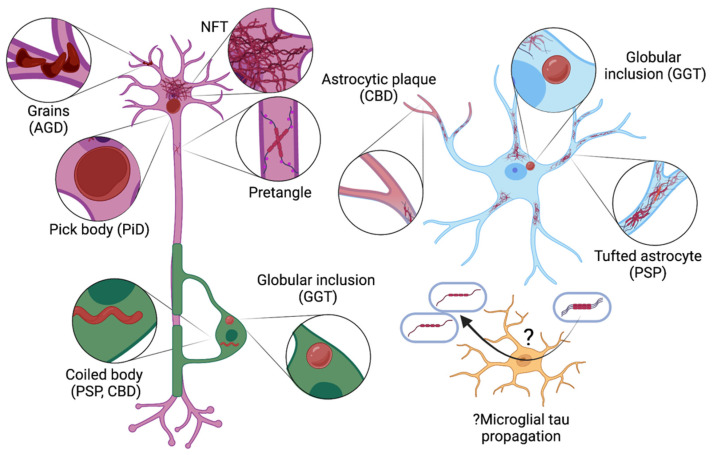
Pathological tau lesions seen in different cell types in tauopathies. In the neuron (purple), tau aggregates include pretangles and neurofibrillary tangles (NFTs), round cytoplasmic inclusions (Pick bodies; typical of Pick disease [PiD]) and grains (dendritic swellings, seen in argyrophilic grain disease [AGD]). Oligodendrocytes (green) may develop tau aggregates in the form of globular inclusions (a feature of globular glial tauopathy [GGT]) or coiled bodies (seen in progressive supranuclear palsy [PSP] and corticobasal degeneration [CBD]). Tau may accumulate in astrocytes (blue) as star-like tufts, plaques, or globular inclusions (hallmarks of PSP, CBD, and GGT, respectively). While pathological tau lesions do not occur in microglia (yellow), they may be implicated in propagating tau (this uncertainty is indicated by the symbol ‘?’).

**Figure 4 ijms-23-07307-f004:**
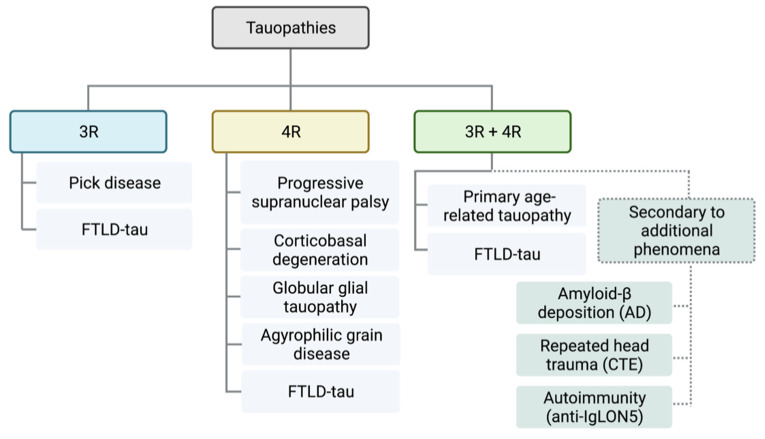
Molecular classification scheme of tauopathies. Tauopathies are divided into those with pathological tau aggregations of isoforms with 3 repeats (3R, blue) or 4 repeats (4R, yellow) of the microtubule binding domain, or a mixture of 3R and 4R isoforms (3R + 4R, green). Secondary tauopathies (broken lines) are associated with additional pathologies or etiologies such as amyloid-β protein deposition (i.e., Alzheimer’s disease (AD)), repeated head trauma (i.e., chronic traumatic encephalopathy (CTE)) or autoimmune disease (e.g., anti-IgLON5-related tauopathy). Primary tauopathies (unbroken lines) are those where tau deposition is the predominant pathology driving the neurodegeneration.

**Table 1 ijms-23-07307-t001:** ELISA techniques to measure CSF total tau.

Antibodies	Publication	CSF Total Tau Increase in AD vs. Control
AT120 capturing antibody + polyclonal detection antibodies (rabbit anti-human tau antiserum)	Vandermeeren 1993 [23]	~100× AD (n = 27) 10.9 ± 4.9 pg/mL Control (n = 51) 0.1 ± 0.5 pg/mL
16B5 capturing antibody + 16G7 detection antibody	Vigo-Pelfrey 1995 [25]	~2× AD (n = 71) 361 ± 166 pg/mL Control (n = 26) 190 ± 80 pg/mL
Microsphere ELISA: polyclonal (anti-ht2) + monoclonal antibodies (F-F11 + F-H5)	Mori 1995 [24]	~2× AD (n = 14) 820 ± 90 pg/mL Control (n = 36) 380 ± 120 pg/mL
AT120 capturing antibody + HT7 detection antibody + BT2 detection antibody	Blennow 1995 [2]	~3× AD (n = 44) 524 ± 280 pg/mL Control (n = 31) 185 ± 50 pg/mL

AD = Alzheimer’s disease; CSF = cerebrospinal fluid; ELISA = enzyme-linked immunosorbent assay; pg/mL = picograms per milliliter.

**Table 2 ijms-23-07307-t002:** Clinical features and tau lesional characteristics of selected tauopathies.

Tauopathy	Isoform	Tau Lesions and Other Pathological Changes by Cell Type			Clinical Features
		Neuron	Astrocyte	Oligodendrocyte	
**Primary tauopathies (FTLD-tau)**					
Pick disease	3R	Pick bodies, ballooned neurons	Ramified astrocytes	Pick-body-like inclusions	Behavior change, social disinhibition, parkinsonism
Progressive supranuclear palsy	4R	Globose NFTs, pretangles	Tufted astrocytes	Coiled bodies	Vertical supranuclear gaze palsy, postural instability with falls
Corticobasal degeneration	4R	Ballooned neurons, pretangles	Astrocytic plaques	Coiled bodies	Asymmetric limb apraxia and rigidity, executive dysfunction
Globular glial tauopathy	4R	Pretangles	Globular inclusions	Globular inclusions	Highly variable: may include behavior change, parkinsonism, cognitive impairment
Argyrophilic grain disease	4R	Grains, ballooned neurons	Ramified astrocytes	Coiled bodies	Personality change, emotional dysregulation, memory impairment
Primary age related tauopathy	3R + 4R	NFTs	N/A	N/A	Cognitive impairment
**Secondary tauopathies**					
Alzheimer’s disease	3R + 4R	NFTs	N/A	N/A	Memory loss, other cognitive dysfunction
Chronic traumatic encephalopathy	3R + 4R	NFTs	Thorn-shaped astrocytes	N/A	Memory loss, personality change, motor decline
Anti-IgLON5-related tauopathy	3R + 4R	NFTs	N/A	N/A	Sleep apnea, non-REM sleep parasomnias, bulbar dysfunction

## Data Availability

Not applicable.

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
