# Peer review of "Tau as a Biomarker of Neurodegeneration"

_ijms, 2022, doi:10.3390/ijms23137307_

Round 1
Reviewer 1 Report
This is a well-written and timely review, which focuses on the use of tau as a biomarker for Alzheimer’s disease and related disorders (AD). As clearly indicated by the authors, tau has become fundamental for differentiating the different kinds of tauopathies. The historic perspective sets this review apart from others. The authors describe and cite all the major publications. The figures are of quality and further raise the quality of the review. Overall, this is one of the best reviews I read over the last five years.
I only have two minor comments, which might improve an already great review:
1. The authors could mention the hypothesis that some tau toxicity may come from an imbalance between the 3R/4R ratio.
2. To my surprise, the authors did not discuss a potential role for tau oligomers in tauopathies.
Reviewer 2 Report
The Review manuscript by Sarah Holper et al., entitled "Tau as a Biomarker of Neurodegeneration" was well written and well presented for publication. The authors highlight and update the molecular and technological advances and clinical applications of tau as a biomarker of neurodegeneration. The manuscript is scientifically sound and contributes significantly to the field of neuroscience and Diagnostics, and Therapeutics of neurodegenerative diseases.
